# Negative emotional reactions to criticism: Perceived criticism and source affects extent of hurt and relational distancing

**Michelle Jin Yee Neoh**[1], **Jia Hui Teng**[1], **Albert Lee**[1], **Peipei Setoh**[1], **Claudio Mulatti**[2], **Gianluca Esposito**[1,2,3] *

**1** Psychology Program, School of Social Sciences, Nanyang Technological University, Singapore, Singapore, **2** Department of Psychology and Cognitive Science, University of Trento, Rovereto, Italy, **3** Lee Kong Chian School of Medicine, Nanyang Technological University, Singapore, Singapore

* gianluca.esposito@ntu.edu.sg

**Data Availability Statement:** The dataset generated during the current study is available in the open access institutional data repository (DR-

## Abstract

Criticism is commonly perceived as hurtful and individuals may respond differently to criticism originating from different sources. However, the influence of an individual's perception of criticism in their social relationships on negative emotional reactions to criticism has not been examined across different relational contexts. The present study investigated the influence of perceived criticism and relational contexts–mother, father, romantic partner, and workplace supervisor–on the feelings of hurt and relational distancing experienced upon receiving criticism. Participants (N = 178) completed the Perceived Criticism Measure and read vignettes describing scenarios of personally directed criticism in the four relational contexts. Significant main effects of perceived criticism and source were found on levels of relational distancing. Participants who perceived their relational partner to be more critical experienced greater distancing upon receiving criticism from them. Greater relational distancing was experienced for criticism received from workplace supervisors compared to mothers, fathers and romantic partners. Results indicate that emotional reactions and relationship outcomes in response to criticism can differ based on individual differences and relational context, suggesting their role in relationship maintenance and development of psychopathology.

## Introduction

Criticism is defined as negative evaluative feedback received from other people in social interactions [1, 2]. It can be construed by recipients as either a positive or distressing experience, where much of the impact of criticism is contingent on the attributions the individual makes about the criticism [3–5]. Receiving social evaluations such as social rejection in the form of negative feedback can result in negative self-evaluations [6] and social pain [7]. Criticism has been reported to be involved in majority of descriptions of hurtful events [8] and makes up a category of hurtful interactions in romantic relationships [9]. On the other hand, positive attributions can also be made about criticism such as constructive criticism, which can be interpreted as promoting understanding and providing direction for improvement [10]. Hence, the

NTU) at the link: https://doi.org/10.21979/N9/R9AFNM.

**Funding:** GE and CM were supported by a grant from the Italian Ministry of University and Research Excellence Department Grant (2018-2022) awarded to the Department of Psychology and Cognitive Science, University of Trento, Italy The funders had no role in the study design, data collection, and analysis, decision to publish, or preparation of the manuscript.

**Competing interests:** The authors have declared that no competing interests exist.

investigation of individual differences in the perception of criticism in interpersonal relationships is important in providing insight to understanding the negative consequences of criticism on relationships.

## Hurtful communication in relationships

Previous research has looked into the factors affecting the responses to hurtful communication, such as feelings of hurt elicited, among other consequences. Firstly, the effects of hurtful behaviour on a relationship are influenced by the intensity and frequency that they are experienced in a relationship. How often people believe they have been hurt has been linked to relationship outcomes and their response to such behaviour. When messages are perceived to be a continuing pattern of hurtful communication, they resulted in increased relational distancing and more intense social pain experienced [11]. Secondly, research has shown that perceived intentionality behind hurtful communication may affect the way people explain or interpret it. Negative attributions of criticism have been associated with greater upset due to criticism whereas positive attributions were associated with lesser upset [10]. The association between perceived intention and resulting emotions is moderated by the perceived frequency of hurtful experiences. Specifically, perceptions of intentionality are associated with greater feelings of hurt when individuals believe their partners do not usually hurt them as opposed to individuals who believe they are often hurt by their partner [11]. It appears that an individual's perception of the frequency and intentionality of hurtful communication plays an important role in the resulting experience of hurt and relational distancing. Hence, it can be expected that an individual who has a greater tendency to perceive criticism in their interpersonal relationships would experience greater feelings of hurt in response to criticism.

One measure of the perception of criticism in one's relationships is the Perceived Criticism Measure (PC) [12], which reflects the amount of criticism, in an individual's closest or most meaningful relationships, such as with a romantic partner, spouse, or parent and shows a strong correlation with perceptions of destructive criticism [13]. The PC construct represents both objective and subjective experiences of criticism [14], relating to objective levels of criticism in the social environment and the individual's perception of how critical a relational partner is of them. Neuroimaging studies have found differences in neural activity in response to criticism between individuals with high versus low PC. Individuals who rate their relationships as high in PC showed differences in activation in regions associated with emotion reactivity and regulation [15] and show increased activation in regions associated with cognitive control and emotion regulation in response to criticism involving romantic partners and parents but decreased activation in these areas in response to criticism involving friends [16]. Since increased levels of hurt and relational distancing are experienced in ongoing patterns of hurtful communication, this suggests that individuals who rate their target relationship as high in PC may be more likely to experience higher levels of hurt and relational distancing in response to criticism.

## Criticism in different relational contexts

Previous studies have considered how relationships' emotional contexts contribute to the interpretations of hurtful behaviour, including hurtful communication. Specifically, the emotional context characterising family environments is linked to the experience of hurtful behaviours. The perception of the family context as aggressive tended to lead to perceptions of hurtful behaviours by family members as intentional while the perception of the family environment as lacking in affection led to experiences of less emotional pain from hurtful episodes, suggesting habituation to situations of hurtful interactions and numbing to hurtful feelings or

normalisation of such behaviours in the family environment [17]. This indicates that emotional responses to hurtful behaviour may not be consistent over time. Consistent or repeated exposure to criticism could lead to changes in cognitive and emotional responses to criticism through sensitisation and/or habituation. Firstly, sensitisation suggests that an increased sensitivity to emotional pain may result from repeated exposure where exaggerated emotional responses are proposed to be observed over time [18]. Romantic partners, family members, and supervisors were identified as generating the strongest negative feelings [19], highlighting the need to investigate how these relational contexts influence the experience of hurt in response to criticism. In the context of criticism, frequent exposure to criticism in one's relationships may lead to a greater likelihood of perceiving criticism and/or a lower threshold for being hurt by criticism. Secondly, a habituation model proposes that repeated exposure may result in emotional numbness to feelings of hurt where the original emotive response of an individual decreases gradually with time or repeated exposure [20]. The perception of frequent and intentional hurt by relational partners led to less intense feelings of hurt, suggesting the development of "emotional calluses" [11]. Similarly, individuals have been observed to get used to emotional pain and become less sensitive to their own hurt feelings with repeated, ongoing exposure to certain hurtful behaviours [17]. Hence, these models suggest that the prevalence of criticism in a relationship can influence how individuals interpret hurtful communication such as criticism from others and their emotional response.

Mental representations of one's relationships with others are expected to inform an individual's behaviour while being used to predict and interpret others' behaviours [21], suggesting that the same criticism may elicit different reactions when originating from different sources. The relationship of PC with depressive symptoms also differs across sources where only PC ratings for family members and romantic partners who lived with the respondents significantly predicted change in depressive symptoms but PC ratings for friends and individuals ranked most influential did not [22]. This finding suggests that the impact of perceived criticism on an individual may differ depending on the environment in which the relationship is embedded in. In addition, individuals who are depressed or maritally discordant have also been found to display a criticality bias–a tendency to over-perceive criticism–which was found to be associated with marital attributions of behaviour [23], suggesting that such bias appears to be indicative of the views an individual holds of their spouse and marriage.

Previous studies have found that hurtful behaviour by family members tend to elicit greater emotional pain than hurtful behaviour by others in non-familial and non-romantic relationships, while a lower distancing effect on the relationship was observed by people who reported hurt feelings by a family member compared to other relational partners [24]. It has been suggested that extensive knowledge that family members possess enables them to be particularly skilled at hurting one another. The value and permanence of familial relationships may also contribute to greater hurt experienced. Two reasons have been proposed for the lower likelihood of relational distancing by family members. Firstly, a family bond is involuntary and permanent nature compared to other relationships such as romantic relationships and friendships which are formed by choice. Secondly, considering the amount of time spent with family, the common shared history may hold greater importance and significance on relational outcomes than a single, particular hurtful episode. In line with these proposed reasons, hurtful messages from romantic partners have the ability to cause as much hurt as family members but the impact of hurtful messages on the romantic relationship is greater [24]. Such findings on the differing impact of hurtful behaviours across relational contexts align with the idea that close and salient relationships are incorporated in the self-concept [25]. Hurt feelings can also be elicited by someone they do not know well such as acquaintances or strangers [26]. Hence, it is highly likely that hurt and relational distancing experienced will differ as a function of the

relational context. The present study investigates familial, romantic and workplace relationships as ties with parents and romantic partners are the most crucial [27], and individuals spend a significant amount of time at work, where criticism is commonly encountered from supervisors.

**Significance and aim of study.** A limited number of studies have investigated sensitivity to criticism where criticism sensitivity was found to exhibit convergent validity with measures of upset in response to criticism [28] but not with perceptions of criticism [28, 29]. However, previous studies have not looked specifically at (i) how individual differences in the perception and response to criticism influence experiences of hurt and relational distancing as a result of criticism and (ii) whether emotional sensitization or habituation occurs in response to criticism in various interpersonal relationships. Few studies have also investigated the PC construct in Asian contexts [e.g. 30, 31] although cultural differences in the perception and attributions of criticism have also been found [e.g. 10, 31]. Previous research indicated (i) correlations between patients' perceived criticism and observer ratings of criticism from relatives only in White patients but not Black patients [32], (ii) observer ratings of criticism predicted relapse and poor clinical outcomes only in White but not Black participants and (iii) associations between perceived criticism and poor outcomes in both groups [33, 34]. More specifically, Black participants in a community sample reported more positive attributions but perceived greater destructive criticism compared to White participants [3]. Allred & Chambless [10] also found that Black participants were significantly less upset by perceived criticism from relatives compared to White participants and perceptions of relatives' warmth were only observed to be correlated with less upset for Black participants and not White participants.

Given the well-established empirical association between excessive criticism and levels of PC with psychopathology [14, 35; see 14 for a review] and findings that ratings of emotional upset in response to relatives' criticism predicted depressive and manic symptoms in bipolar patients [36], studying how PC relates to feelings of hurt and relational distancing can provide insight into the relationship between an individual's perception of criticism and the consequences on emotions and the relationship. Hence, the present study aims to investigate the relationship between individual differences in PC and experiences of hurt and relational distancing in response to criticism in four different relational contexts: (i) romantic partners, (ii) mothers, (iii) fathers, and (iv) workplace supervisors in a Singaporean sample.

In line with previous findings on the association between PC and (i) increased upset [10, 36, 37] and (ii) lower relationship and marital satisfaction [23, 38], we formulated the following hypotheses:

Hypothesis 1: Individuals who have higher perceived criticism ratings of their relational partner would experience higher levels of hurt and relational distancing in response to criticism compared to individuals who have lower perceived criticism ratings.

Hypothesis 2: Higher levels of hurt and lower relational distancing will be experienced in familial relationships compared to other social relationships.

## Methodology

### Participant recruitment

Participants were recruited in Singapore ($N$ = 178, male = 83, female = 95, $M_{age}$ = 21.3, $SD_{age}$ = 2.23) through (i) a psychology undergraduate course and compensated with course credits and (ii) advertisements and compensated with remuneration (Table 1). The study was approved by the Psychology Ethics Committee at Nanyang Technological University (PSY-IRB-2020-007). Written informed consent was obtained from participants before completing the

**Table 1. Descriptive statistics for age, gender, relationship status, and past work experience.**

| Age | |
| --- | --- |
| Mean | 21.3 |
| SD | 2.23 |
| **Gender** | |
| Male | 83 |
| Female | 95 |
| **Relationship status** | |
| Currently in a romantic relationship | 47 |
| Have previously been in a romantic relationship | 50 |
| Never been in a romantic relationship | 81 |
| **Work experience** | |
| Currently working | 49 |
| Past work experience | 121 |
| No work experience | 8 |

questionnaire. All participants answered questions regarding their demographic information and the following questionnaire measures hosted on Qualtrics.

## Questionnaire measures

Perceived criticism ratings were obtained from each participant for their romantic partner, mother, father and workplace supervisor. PC was assessed with the question "How critical is (the relative/workplace supervisor) of you?" rated on a 10-point scale [12]. A high rating indicates a high amount of criticism "[getting] through" to the individual in the particular relationship being rated. PC ratings have demonstrated high predictive validity, correlated with expressed emotion and high test-retest reliability [12, 39].

Relationship quality was measured using questions adapted from the Quality of Marriage Index [40]. Participants rated the quality of each of the relationship type: romantic partner, mother, father and workplace supervisor. There were 5 items rated on a 7-point Likert scale and 1 item rated on a 10-point Likert scale where higher scores reflected higher relationship quality.

## Experimental procedure

Four vignettes describing hypothetical scenarios involving criticism were constructed. All vignettes were approximately 120 words, consisting of a brief paragraph describing the background of the event precipitating the criticism and a block quote of the criticism received from one of the four different sources. The vignettes were written in the first-person perspective to increase identification with the protagonist described in the vignettes by participants.

The four vignettes were shown in the same order to participants. Participants were randomly assigned to conditions which differed in terms of the order of the relationship type being described in each vignette. The criticism in each vignette was described to originate from either a romantic partner, mother, father or workplace supervisor. The order of the relationship types being described in the vignettes are as follows: (a) Mother-Father-Partner-Supervisor, (b) Father-Supervisor-Mother-Partner, (c) Supervisor-Partner-Father-Mother and (d) Partner-Mother-Supervisor-Father. After reading each vignette, participants were asked to rate the extent of hurt and relational distancing they would experience if they were in the hypothetical situation described on a 5-point scale (1 being not at all and 5 being completely).

### Analytic plan

To test the hypotheses, two-way, mixed analysis of covariance was conducted in order to investigate the effect of PC and source of criticism on (i) level of hurt and (ii) relational distancing. Participants were grouped into high or low on PC through a median split ($Mdn_{Mother}$ = 4, $Mdn_{Father}$, $Mdn_{Partner}$, $Mdn_{Supervisor}$ = 5). The medians in this sample are also similar to previous studies on PC which have also used median splits (e.g. [12, 41]) to facilitate interpretation of results. Relationship quality was included in the analysis as a covariate. For (ii), 1 participant was omitted due to missing data. For participants who indicated that they (i) have never been in a romantic relationship and/or (ii) did not have previous work experience, their ratings for the vignettes involving criticism from (i) romantic partners and (ii) supervisors respectively were not included in the data analysis.

## Results

### Preliminary analysis

PC ratings ranged from 1 to 10 (mean = 4.7, SD = 2.78). The correlations between PC ratings and ratings of hurt ($r$ = 0.003) and relational distancing ($r$ = 0.065) were not significant. The correlation between ratings of hurt and relational distancing was significant ($r$ = 0.506, $p$ = < .001).

We conducted independent samples t-test to check whether there were differences between the participants recruited from the university and those who were not. There were no significant differences between the two groups in terms of (i) ratings of hurt ($t_{(621)}$ = 0.19, $p$ = 0.85) and (ii) ratings of relational distancing ($t_{(617)}$ = -0.09, $p$ = 0.93). Hence, both groups were analysed together in the main analyses.

### Perceived criticism and source of criticism on hurt

Table 2 summarises the analysis of covariance table of PC group and Source on hurt. Table 4 summarises the means of hurt grouped by Source. The (i) main effect of PC group ($F_{(1,170)}$ = 0.69, $p$ = 0.41), (ii) main effect of Source ($F_{(3,437)}$ = 2.37, $p$ = 0.07) and (iii) interaction effect of PC group x Source ($F_{(3,437)}$ = 1.15, $p$ = 0.33) were not significant.

### Perceived criticism and source of criticism on relational distancing

Table 3 summarises the analysis of covariance table of PC group and Source on relational distancing. Table 4 summarises the means of relational distancing grouped by Source. A significant main effect of PC group was found ($F_{(1,169)}$ = 8.22, $p$ = 0.005, $\eta^2$ = 0.007) such that greater relational distancing was experienced for the high PC group (Fig 1). A significant main effect of Source was found ($F_{(3,434)}$ = 13.37, $p$ < .001, $\eta^2$ = 0.046). The interaction effect of PC group

**Table 2. Analysis of variance table for levels of hurt.**

|  | SS | df | MS | F | p-value |
|---|---|---|---|---|---|
| Perceived Criticism Group | 1.7 | 1 | 1.68 | 0.69 | 0.41 |
| Source | 7.5 | 3 | 2.49 | 2.37 | 0.07 |
| Perceived Criticism Group x Source | 3 | 3 | 1.22 | 1.15 | 0.33 |
| Relationship quality | 3.8 | 1 | 3.83 | 1.59 | 0.21 |
| Error | 460.2 | 437 | 1.05 |  |  |

$^*$ $p$ < .05, $^{**}$ $p$ < .01, $^{***}$ $p$ < .001

**Table 3. Analysis of variance table for levels of relational distancing.**

| | SS | df | MS | F | p-value |
|---|---|---|---|---|---|
| Perceived Criticism Group | 19.0 | 1 | 19.05 | 8.22 | 0.005** |
| Source | 50.5 | 3 | 16.82 | 13.37 | < .001*** |
| Perceived Criticism Group x Source | 4.8 | 3 | 1.61 | 1.28 | 0.28 |
| Relationship quality | 7.8 | 1 | 7.79 | 3.36 | 0.07 |
| Error | 545.8 | 434 | 1.26 | | |

* $p < .05$,

** $p < .01$,

*** $p < .001$

x Source was not significant ($F_{(3,434)} = 1.28$, $p = 0.28$). Table 5 summarises the post-hoc pairwise comparisons on the marginal means of relational distancing by Source. Significantly lower relational distancing was experienced for vignettes describing criticism from (i) mothers ($t = -5.38$, $p < .001$, Bonferroni corrected), (ii) fathers ($t = -4.38$, $p < .001$, Bonferroni corrected) and (iii) partners ($t = -2.91$, $p < .001$, Bonferroni corrected), compared to workplace supervisors (Fig 2).

## Discussion

The present study aimed to investigate the relationship between perceptions *of* criticism from relationship partners in influencing the response to criticism in different relational contexts in terms of the experiences of hurt and relational distancing.

Firstly, results in the present study partially supported Hypothesis 1 where participants in the high PC group showed higher levels of relational distancing compared to those in the low PC group but not higher levels of hurt. The finding that the high PC group did not report feeling higher levels of hurt is consistent with previous findings that PC was not significantly correlated with criticism sensitivity, which includes emotional sensitivity–measured by how upset an individual is by criticism [29]. These findings appear at first glance to differ from those in previous neuroimaging results discussed earlier [15]. One possible interpretation of these findings could be that participants with high PC employed relational distancing as a coping mechanism by disengaging themselves from the hurt and social pain associated with criticism. This possibility is consistent with past work on how individuals may use of ego-moving perspective of time, which enables people to psychologically distance themselves from unpleasant, threatening past experiences [42]. Since rejection from close others tends to hurt more than rejection from strangers [8, 26], it could be that a sense of increased distance from one's relational partners can reduce the intensity of hurt experienced in response to hurtful behaviour such as criticism. Hence, it is possible that participants in the high PC group tended to distance themselves from the relationship and in doing so, "distanced" their emotions from the situation and did not show significantly higher levels of hurt feelings compared to participants in the low PC group. In terms of the neural activation patterns found in previous studies, such a coping

**Table 4. Mean (SD) ratings of hurt and relational distancing by source.**

| | Romantic partner | Father | Mother | Supervisor |
|---|---|---|---|---|
| Hurt | 3.31(1.09) | 3.05(1.26) | 2.94(1.25) | 3.17(1.19) |
| Relational distancing | 2.46(1.31) | 2.47(1.31) | 2.33(1.22) | 3.07(1.25) |

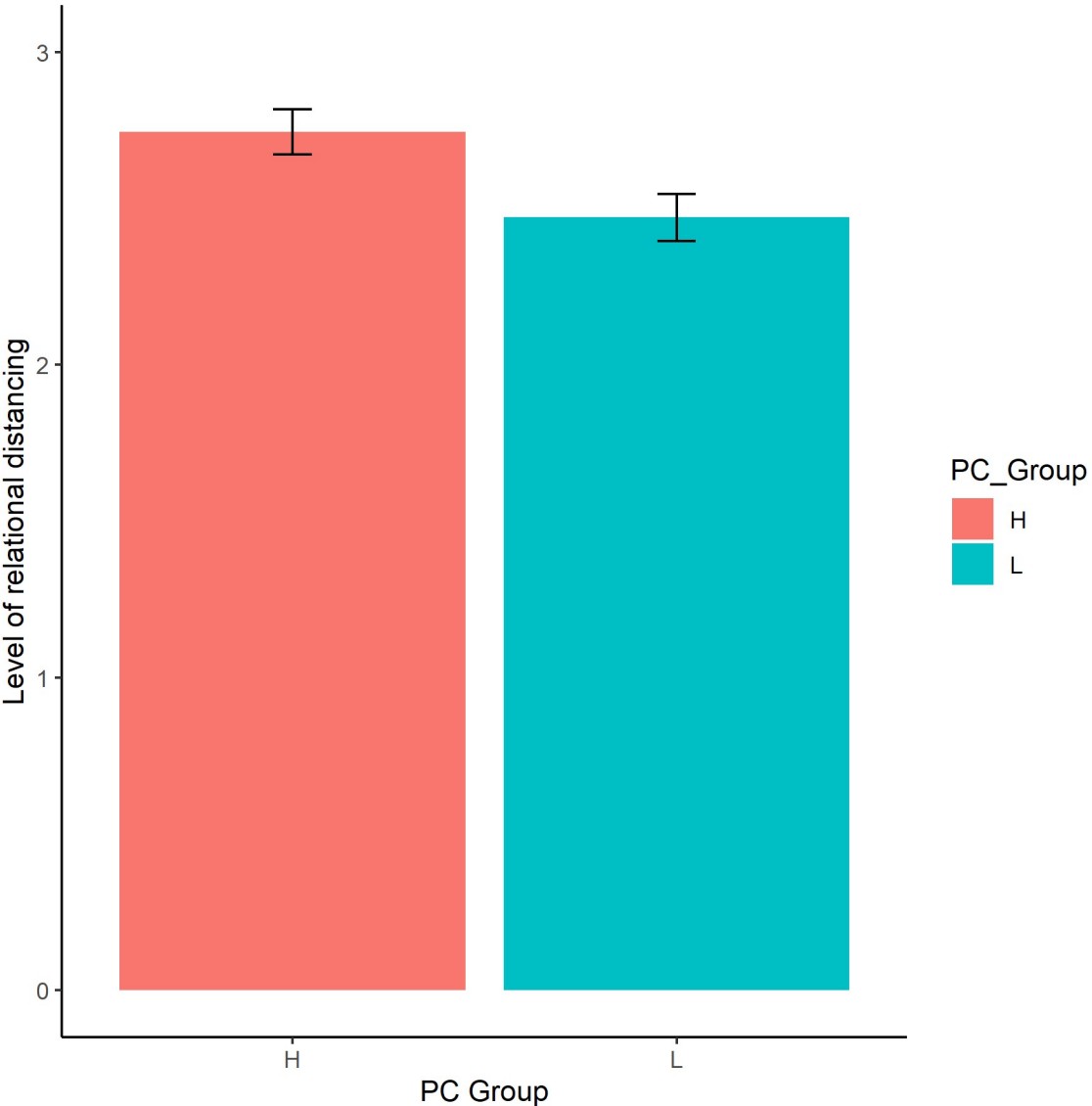

**Fig 1. Plot of means of relational distancing by perceived criticism group.** Note. (i) H: High, L: Low.

mechanism where individuals distance themselves from the relationship and the associated emotions resulting from the criticism may be reflected in the decreased activity previously observed in cognitive control networks as less attention and cognitive resources are being placed on processing the criticism. It is possible that as Hooley et al. [15] found in terms of increased amygdala activity, participants in the high PC group may have experienced more hurt towards the criticism described in the vignettes but did not rate their levels of hurt as significantly higher than participants due to the employment of relational distancing in order to cope with the hurt that they experienced. Another possible interpretation is that participants who tend to perceive higher levels of criticism in a relationship showed emotional habituation towards the experience of hurt arising from criticism, such that the more a participant perceives a relationship counterpart to be critical of them, the more accustomed they were to criticism from these relational partners. This could lead to less hurt experienced with repeated encounters involving criticism. On the other hand, previous findings have indicated that PC

**Table 5. Pairwise comparisons of levels of relational distancing by source for high and low perceived criticism groups.**

| Variable | Pairwise comparison | $t$ | $p$-value (Bonferroni corrected) |
|---|---|---|---|
| Source | Father-Mother | 1.01 | 1 |
| | Father-Partner | -0.78 | 1 |
| | Father-Supervisor | -4.38 | < .001*** |
| | Mother-Partner | -1.62 | 0.63 |
| | Mother-Supervisor | -5.38 | < .001*** |
| | Partner-Supervisor | -2.91 | 0.02* |

\* $p < .05$,

\*\* $p < .01$,

\*\*\* $p < .001$

reliably predicts clinical outcomes [see 14 for review] and greater subjective distress experienced by bipolar disorder patients in response to familial criticism predicted the severity of depressive and manic symptoms [36]. A possible explanation for present findings in the context of previous work could be the association between attributions for the criticism and PC. Chambless et al. [5] found that higher scores for negative attributions predicted higher PC ratings while Peterson et al. [43] found similar associations between PC ratings and the cause and responsibility ascribed to partners' behaviours. It is possible that despite habituating to the experience of hurt from their partners, high PC individuals are more likely to perceive "more" criticism from their partners as well as form negative attributions about such criticisms along with a majority of their partners' behaviours in general. It could be that these high PC individuals experience higher levels of hurt from a consistent perception of multiple events of negative behaviour from their relationship partners as opposed to significantly higher levels of hurt from a single event, compared to low PC individuals who may be less likely to perceive and explain behaviours from their relationship partners as negative. A more consistent perception of negative behaviours from relationship partners could then contribute to the degradation of their relationships with their partners and consequently, to poor clinical outcomes as well. In addition, greater relational distancing observed in participants with high PC also suggest a possible pathway through which PC ratings predict clinical outcomes. By distancing themselves from the relationship partner, high PC individuals may experience greater social isolation or be less likely to seek social support from these relationships, leading to reduced relationship satisfaction which could consequently influence clinical outcomes. Further research can be conducted in order to link the experience of relational distancing with relationship outcomes and clinical outcomes. Future studies could also investigate relationship closeness and how it moderates the relationship between PC and experiences of hurt and relational distancing in response to criticism.

Secondly, results in the present study only partially supported Hypothesis 2, where we hypothesised that higher levels of hurt and relational distancing would be observed in familial relationships compared to other social relationships. However, in the present study, only significant differences in levels of relational distancing but not hurt were observed across different relationships. Results suggest that hurt feelings may not differ in response to criticism originating from the different relationships that were examined in this study. Rather, findings suggest that criticism can be hurtful as long as the individual receiving it perceives it to be. Given that the present experimental design was limited to four scenarios, it may be that source of criticism does not significantly influence the emotional reaction in the context of the

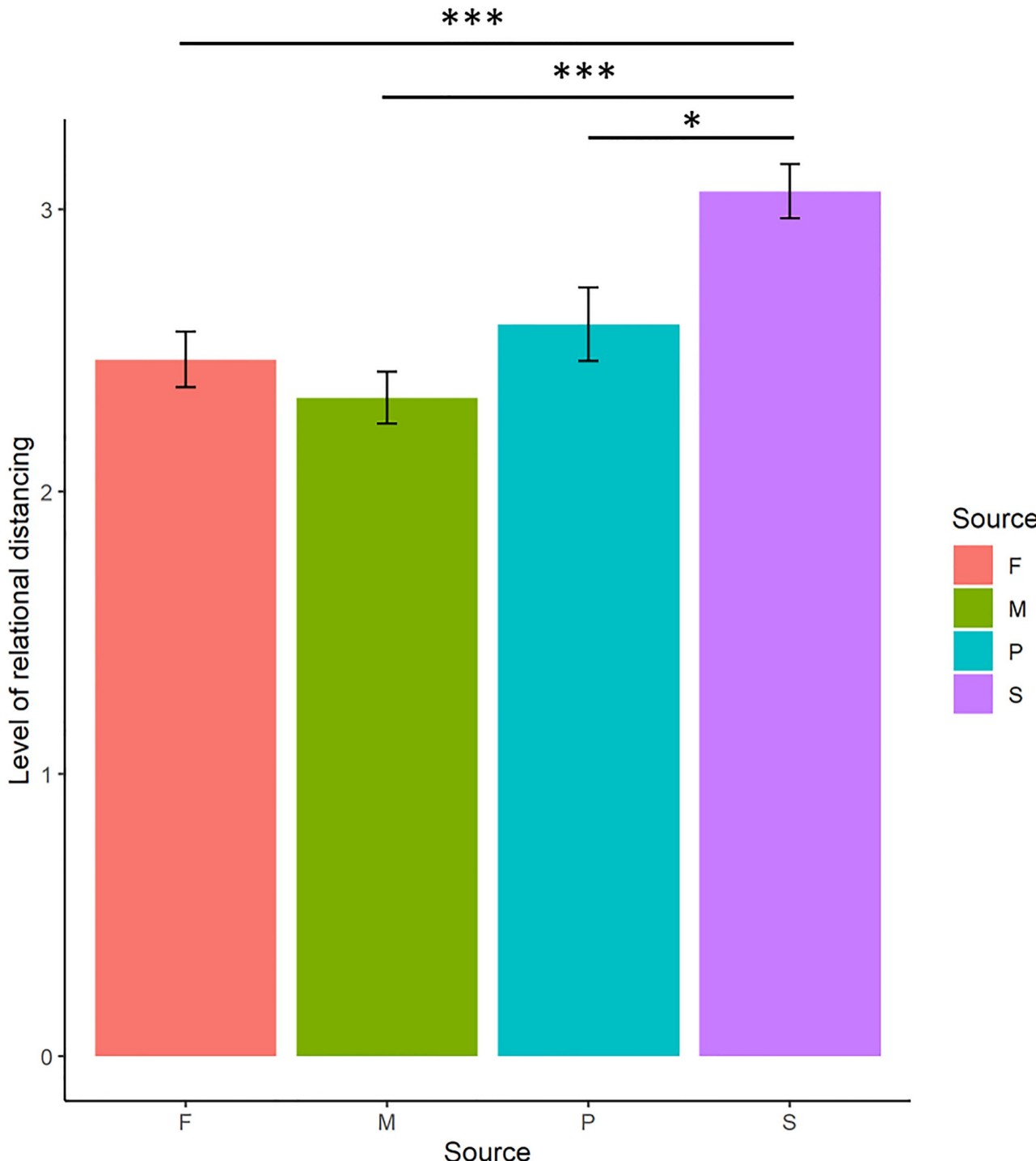

**Fig 2. Plot of means of relational distancing by source.** Note. (i) F: Father, (ii) M: Mother, (iii) P: Partner and (iv) S: Supervisor.

criticism illustrated in the experimental vignettes. Future studies can investigate whether specific content of various criticism interacts with the source of criticism in influencing the emotional reaction and experiences of hurt. Another possible explanation for the finding on similar levels of hurt across relational contexts could relate to the emotion regulation strategies

employed. Previous studies have found that emotion regulation strategies employed differed depending on the relational context where suppression was used more in interactions involving non-close others [44]. Suppression may also be more likely when individuals are motivated to avoid straining the relationship with parents [45] while expression of negative emotions may be more likely in order to ward off excessive parental control [46]. Participants may have been motivated to avoid strain in these relationships–romantic partners, parents and supervisors–and thus, experiences of hurt tended to be suppressed. Since maintenance of interpersonal harmony and respect for authority and seniority are characteristic of the collectivistic culture in Singapore [47], relationship maintenance may be a key motivation for participants in these close relationships as well as the relationship with their supervisor, who is a figure of authority in the workplace and also holds influence over their livelihood and work environment. Future studies can look at the specific emotion regulation strategies employed in response to criticism in social interactions and their effect on the emotional response.

On the other hand, relational distancing was significantly higher in response to criticism from supervisors compared to criticism from parents and romantic partners. This finding aligns with previous work indicating that lower levels of relational distancing tend to be experienced in response to hurtful behaviour from familial relationships [19]. It has previously been suggested that the relatively permanent nature of the family bond can influence and better withstand the experience of hurtful behaviour, which could serve as a possible explanation for present findings. Significantly lower relational distancing was also observed in romantic relationships compared to workplace supervisors, suggesting that the closeness of the romantic relationship may also buffer against distancing experienced in response to criticism. Hence, results suggest that temporary or relationships by choice are more susceptible to degradation as a result of a criticism as opposed to familial relationships. Another possible factor that can contribute to this finding is differences in the attributions of the cause of the criticism in different types of relationships. During conflict, individuals make negative attributions about traits of the other person even with information of the situation causing the behaviour [48]. Individuals were also more likely to make situational attributions for close others' behaviours but not non-close others [49]. In addition, previous work has found an association between attributions and perceptions of criticism. Specifically, higher negative attribution scores were related to higher PC ratings [3, 5], indicating that attributions are related to how much criticism is perceived in a relationship. Similarly, Peterson et al. [43] found that self-reports of causal and responsibility attributions for negative spousal behaviour were related to all types of criticism, suggesting that the attributions individuals make about another's behaviour influence whether the behaviour will be perceived or interpreted to be criticism. Hence, it is possible that individuals may be are more likely to make negative attributions such as being more likely to attribute the cause of the negative feedback received to negative dispositions of the supervisor as opposed to family members, with whom individuals have a longer shared history and knowledge of and a more permanent relationship.

Lastly, greater experience of relational distancing towards supervisors but not for parents supports suggestions that decreased activity in brain areas involved in social cognitive processing towards maternal criticism to protect relationships with parents [50]. It may be that individuals disengage more from criticism from parents and romantic partners than supervisors, as there is greater motivation to protect their feelings and close relationships.

## Implications

Excessive criticism has been associated with negative individual outcomes and the development and recurrence of psychopathology [14]. Present findings suggest that individuals high

in PC may be more vulnerable to the development of psychopathology due to an increased distance from close relationships in response to criticism. For example, increased relational distancing may lead to lower perceived social support or willingness to approach relationship partners for support.

Destructive conflict behaviours including criticism have been found to predict higher divorce rates [51] with longitudinal data following couples up to 14 years indicating the presence of criticism, defensiveness, contempt and stonewalling to be predictive of divorce [52]. Hence, present findings suggest relational distancing as a possible pathway through which criticism can lead to dissolution of marriage. Clinicians can consider individual PC as an aspect in the spousal relationship to be addressed during marital therapy for couples experiencing marriage difficulties and communication problems.

Criticism and feedback are crucial aspects in organisational settings where employees stand to learn and improve one's thinking or task performance. However, negative feedback was found to evoke defensiveness, anger and repudiation of feedback in an organisational setting [53]. Perception of feedback as destructive criticism can also lead to feelings of anger or tension and lower goals and self-efficacy [54]. Findings suggest that considerations of perceived criticism are important for workplace supervisors in building healthy relationships and motivating employees.

## Limitations and future directions

There are a few limitations to our study. First, the study was conducted in healthy individuals. Previous studies have found differences in neural processing of maternal criticism between healthy youth and youth recovered from depression [55], suggesting that experiences of hurt and relational distancing could differ between healthy youth and those with mental health disorders. Such differences could be a possible explanation for the vulnerability to depression. Hence, future studies can replicate the methodology in looking at clinical populations to better elucidate differences in the emotional and behavioural responses to criticism that may influence vulnerability to psychopathology. In addition, a large proportion of the sample in the present study had never been in a romantic relationship. Hence, future studies can look further into criticism occurring in romantic relationships such as criticism between romantic partners as well as spouses. The analysis of the present study was also assumed relationship closeness based on the relationship type where familial and intimate relationships were assumed to be close compared to the relationship with a supervisor. Future studies can include measures of relationship closeness to examine whether experiences of relationship closeness also play a role in the response towards criticism originating in different relationship types.

Second, there are cultural differences in communication styles and emotion expression, which can relate to differences in experiences of hurt and relational distancing in response to criticism. Individualistic cultures tend to have a *low-context* communication style, where assertive behaviour is representative of efficacy and competence whereas collectivistic cultures tend to have a *high-context* communication style, where maintenance of face and regard for interpersonal harmony reflect competence [56–58]. Expression of hurt feelings may be curtailed in collectivist cultures to preserve group harmony or conform with group values whereas individualistic cultures tend to promote self-expression and expression of feelings [47]. Collectivistic cultures also place a greater emphasis on hierarchy and status, suggesting that the extent of hurt feelings and relational distancing may differ in response to sources of criticism in positions of higher status, such as parents and workplace supervisors. In addition, collectivistic cultures tend to employ emotion suppression as an emotion regulation strategy as it minimises the risk of disrupting group harmony whereas individualistic cultures tend to use cognitive

reappraisal [59]. As a result of these cultural tendencies, in the context of Singapore where the present study was conducted, criticism from figures of authority such as supervisors or parents may be more common and participants may be less likely to feel hurt or distanced in response to criticism occurring in these relationships. As discussed in the introduction, perceptions and attributions of criticism have been found to differ across cultures [3, 10, 31]. Hence, similar to these findings in [3, 10], cultural differences in the attributions and perceptions of warmth may be a possible explanation for the findings in the present study between PC ratings and feelings of hurt and relational distancing.

## Conclusion

Hurt feelings are common in interpersonal relationships and further research can provide insight into understanding conditions that moderate the extent of hurt experienced. Findings showed that the source of criticism can influence its impact on the relationship, providing evidence suggesting that different relationship types have varying vulnerabilities in terms of damage to the relationship due to hurtful communication. In addition, findings suggest that hurt feelings and relational distancing can possibly underlie the empirical association between PC and development and recurrence of psychopathology, providing initial evidence that future studies can build on to investigate pathways involving emotion reactions and relationship outcomes that mediate the association between PC and psychopathology.

## Acknowledgments

All participants are gratefully acknowledged. We would like to acknowledge members of the Social Affective Neuroscience Lab at NTU for their assistance in the completion of this project.

## Author Contributions

**Conceptualization:** Michelle Jin Yee Neoh, Jia Hui Teng, Gianluca Esposito.

**Data curation:** Michelle Jin Yee Neoh, Jia Hui Teng.

**Formal analysis:** Michelle Jin Yee Neoh, Gianluca Esposito.

**Investigation:** Michelle Jin Yee Neoh, Jia Hui Teng.

**Methodology:** Michelle Jin Yee Neoh, Jia Hui Teng, Gianluca Esposito.

**Visualization:** Michelle Jin Yee Neoh, Gianluca Esposito.

**Writing – original draft:** Michelle Jin Yee Neoh, Jia Hui Teng, Albert Lee, Peipei Setoh, Claudio Mulatti, Gianluca Esposito.

**Writing – review & editing:** Michelle Jin Yee Neoh, Jia Hui Teng, Albert Lee, Peipei Setoh, Claudio Mulatti, Gianluca Esposito.

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
