## [Decision Letter · Decision Letter 0]

18 Mar 2022

PONE-D-21-36615Negative emotional reactions to criticism: Perceived criticism and source affects extent of hurt and relational distancingPLOS ONE

Dear Dr. Esposito,

Thank you for submitting your manuscript to PLOS ONE. After careful consideration, we feel that it has merit but does not fully meet PLOS ONE’s publication criteria as it currently stands. Therefore, we invite you to submit a revised version of the manuscript that addresses the points raised during the review process.

We look forward to receiving your revised manuscript.

Kind regards,

Sarah Hope Lincoln

Academic Editor

PLOS ONE

Journal Requirements:

Reviewers' comments:

Reviewer's Responses to Questions

**Comments to the Author**

1. Is the manuscript technically sound, and do the data support the conclusions?

Reviewer #1: Partly

Reviewer #2: Yes

2. Has the statistical analysis been performed appropriately and rigorously? 

Reviewer #1: No

Reviewer #2: Yes

3. Have the authors made all data underlying the findings in their manuscript fully available?

Reviewer #1: Yes

Reviewer #2: Yes

4. Is the manuscript presented in an intelligible fashion and written in standard English?

Reviewer #1: Yes

Reviewer #2: Yes

5. Review Comments to the Author

Reviewer #1: I commend the authors for their attempt to expand our knowledge about PC through the lenses of relational distancing and hurt. In this manuscript, which is mostly clear but sometimes difficult to follow for conceptual and grammatical reasons, the authors demonstrate that people high on PC (identified as high on PC using a median split) indicate that they would feel greater distance from relational partners upon being criticized by them and that relational distancing was greatest in the context of relationships the authors assume to be closer (e.g., family vs. workplace relationships). I generally believe that these findings are interesting. However, the manuscript has a number of significant issues, including problems with the conceptualization of PC and other constructs, the presentation of findings, an unclear approach to selecting and presenting relevant background literature, and analytic limitations. It is possible that many of these concerns could be addressed in a major revision.

My primary concerns surround 1) the conceptualization and interpretation of PC and other constructs; 2) the appropriate selection and representation of background literature; and 3) the analytic approach.

1. Concerns about the conceptualization and interpretation of PC and other constructs

a. The first sentence of the introduction reduces criticism to a necessarily negative experience, yet we know that this is not always the case—criticism can be experienced positively, and much of the impact of criticism may rely on attributions about criticism (see e.g., Allred & Chambless, 2014; Peterson et al., 2009; Chambless et al., 2010). The second sentence also suggests that criticism is necessarily a form of social rejection, which we know is not the case. Oftentimes, critical relatives do not mean to be rejecting but may have an internal locus of control that guides them to believe that criticism will help their family member to change (see Hooley’s 2007 review of expressed emotion). People receiving criticism may also make positive attributions about it (e.g., “my mother is saying this to try and help me”). The authors do consider attributions in the next section, but that body of work needs to inform their introduction paragraph as well.

b. p. 4 lines 74-76. This statement is not exactly supported or in line with how the authors measure criticism. The PCM does NOT actually assess frequency of criticism at all, but rather the global perception of criticism from a relationship partner, which could be influenced by a number of factors which include but which are certainly not limited to frequency of criticism. If the authors want to test the idea that “an individual who has a greater tendency to perceive criticism in their interpersonal relationships would experience greater feelings of hurt in response to criticism,” then they need to use a different measure.

c. p. 4 lines 78-79. The statement that the PCM “usually” reflects “perceptions of destructive criticism” is not a fair characterization of that finding.

d. p. 6 lines 121-126: this paragraph seems like very weak justification for proposing differences across different relationships/sources. The authors may do better to use the PC literature to support the idea that source matters (e.g., it is highly surprising that Renshaw, 2008 is not considered). The material on self-concept does not seem to add much to the author’s presentation of background and feels somewhat disjointed.

e. The authors seem to assume differential levels of closeness between different relationship partners. However, closeness cannot be assumed just based on relationship and should likely have been measured in this study. Do the authors have data on closeness? Or data to speak to whether participants lived with (i.e., were at least physically close to) certain partners (see Renshaw, 2008)?

f. p. 12 line 244: the authors seem to be misusing the term “individual factors” – they did not actually test individual factors (e.g., personality variables, other individual differences) that predict hurt and relational distancing besides hurt and relational distancing.

g. The finding that the high PC group did not report higher levels of hurt is consistent with past work that is not mentioned here (e.g., Masland, Drabu, & Hooley, 2018)

h. The authors note that their findings, which are not “behavioral observations” as they describe them (p. 13 line 249) are discordant with neuroimaging findings. This needs expansion and further explanation/interpretation

i. How does the authors’ interpretations of the possible roles of distancing and habituation (p. 13) square with the decades of research showing that high PC predicts poor clinical outcomes? Or with neuroimaging findings? For example, habituation does not seem consistent with DLPFC findings (Hooley, Siegle, & Gruber, 2012) or the results of cognitive work (Masland et al., 2015)

j. p. 15 line 302: the authors suggest that workplace relationships are different from romantic/family relationships because they are “temporary or… by choice.” However, romantic relationships can also be temporary or by choice. This is a good example of why it is problematic that the authors assumed closeness in certain relationships without actually measuring the degree of closeness

k. p. 15 line 308: the authors again assume closeness although it was not measured. They also do not seem to recognize that attributions about PC vary even in close relationships (e.g., Allred & Chambless, 2014; Peterson et al., 2009; Chambless et al., 2010)

l. Some research has examined emotional upset to criticism, including Miklowitz et al. (2005), who found that emotional upset, rather than criticism, predicted outcomes for bipolar disorder. I am quite surprised that this is not considered given the manuscript’s focus on hurt

m. p. 16 line 336: the authors suggest that their findings show “that considerations of individual differences in how criticism is perceived are important for workplace supervisors in building healthy relationships and motivating employees” yet they did not measure individual differences in how PC is perceived. They measured the influence of PC and source on relational distancing and hurt, not what individual differences might predict PC

n. It is not enough to say that there are cultural differences of note. I would like to see more about how specifically these differences may have limited or influenced the results

2. Appropriate selection and representation of background literature: generally I am concerned that the authors have not adequately tapped into the somewhat small but very rich PC literature. They do not seem to consistently cite the most relevant research.

a. p. 7 line 154: there is a wealth of literature that supports their claim of a well-established link between PC and psychopathology, and it is unclear why they chose the two studies they did in this location, particularly #24. They could cite a review paper here or could cite a broader range of relevant and seminal literature supporting this link (e.g., see papers reviewed in Masland & Hooley, 2015)

b. p. 7 line 150: this may be true, but there has been some limited work on criticism sensitivity that is relevant here (e.g., Masland et al., 2018; White et al., 1998)

c. Where the authors use citation #31 it would be more appropriate to cite work related to attributions and PC (e.g., e.g., Allred & Chambless, 2014; Peterson et al., 2009; Chambless et al., 2010)

d. Citation 19 does not seem to apply necessarily to criticism but more broadly to hurtful interactions. This manuscript would be much better support and contextualized with more reliance on the PC literature

e. The authors do not seem to recognize that attributions about PC vary even in close relationships (e.g., Allred & Chambless, 2014; Peterson et al., 2009; Chambless et al., 2010)

f. Some research has examined emotional upset to criticism, including Miklowitz et al. (2005), who found that emotional upset, rather than criticism, predicted outcomes for bipolar disorder. I am quite surprised that this is not considered given the manuscript’s focus on hurt

3. Analytic approach

a. The link for the data repository appears to be broken

b. Why did the authors use a median split for PC ratings? I am aware that this is very common in the PC literature, but it nevertheless requires justification as it has significant limitations as an analytic strategy. In this case it seems that a dimensional approach is both possible and likely to be more informative

c. There are a number of analyses missing that should be included for complete review of this paper and its findings: the range of PC scores (including the range in each group), the overall PC mean/SD and means/SDs by group, the correlation of PC with distancing and hurt, the correlation of distancing and hurt

d. Although the authors use a categorical analysis approach, they inappropriately use dimensional language to describe their findings (e.g., the abstract reads “the more critical participants perceived the relational partner to be, the more distanced they felt upon receiving criticism from them”)

4. Additional Concerns

a. The authors describe sensitization and habituation models for understanding the impact of criticism. Their hypotheses align with a sensitization model, but they do not give sufficient justification for why they chose the sensitization model over the habituation model

b. On a more granular level, there are times when the manuscript is difficult to follow because of missing words or sentence structures that could be more straightforward. This is a minor concern in the broader context of this review.

Reviewer #2: This is interesting research in a novel area. The paper is generally well written. The statistics are simple but effectively examine the research questions. Some minor changes are recommended below. In particular, some additional statistics are required. Overall, this is valuable research that adds to the current knowledge-base.

Abstract

The first two sentences of abstract are confusing. Please re-phrase

Introduction

Line 153 mentions that PC is linked to psychopathology. It would be beneficial, at some point in the introduction, to briefly mention which psychopathologies/diagnoses specifically are linked to PC to clarify the clinical relevance of examining PC

Method

Information should be included indicating where participants were recruited from (i.e. country)

Results

Were there any differences in results between university-recruited cohort and the other cohort? This should be assessed statistically

Was there any significant difference in results for participants who were working/previously worked vs never working? Significant differences between those who were in/had been in relationship vs never been in relationship? Again analysis is needed to test this.

The inclusion of 8 participants who had never worked is problematic, as they would never have experienced a professional supervisor, as is the inclusion of participants who have never been in a relationship

Discussion

Line 269 “Secondly, results in the present study partially supported Hypothesis 2 where relational distancing was significantly across the different relationships but levels of hurt were not.” This is unclear. Please rephrase.

Line 271 “Results suggest that hurt feelings…” Also unclear, please re-phrased

The discussion includes examination of the possible implication of cultural contexts. This is an important point. However, it is not included at all in the introduction. It would be beneficial to have the review of previous research indicate in which cultural contexts previous research was conducted and greater discussion in the introduction about this issue. Further, the aim of the research should be amended, i.e. aim: to examine the research questions in the context of the Singaporean culture.

A large proportion of the sample had never been in a romantic relationship. Given that one component of the research was about romantic relationships, this is a notable limitation of the research and should be mentioned in the discussion in the ‘limitations’ section.

6. PLOS authors have the option to publish the peer review history of their article (what does this mean?). If published, this will include your full peer review and any attached files.

Reviewer #1: No

Reviewer #2: No

While revising your submission, please upload your figure files to the Preflight Analysis and Conversion Engine (PACE) digital diagnostic tool, https://pacev2.apexcovantage.com/. PACE helps ensure that figures meet PLOS requirements. To use PACE, you must first register as a user. Registration is free. Then, login and navigate to the UPLOAD tab, where you will find detailed instructions on how to use the tool. If you encounter any issues or have any questions when using PACE, please email PLOS at figures@plos.org. Please note that Supporting Information files do not need this step

---

## [Author Response · Author response to Decision Letter 0]

29 Apr 2022

Reviewer #1: I commend the authors for their attempt to expand our knowledge about PC through the lenses of relational distancing and hurt. In this manuscript, which is mostly clear but sometimes difficult to follow for conceptual and grammatical reasons, the authors demonstrate that people high on PC (identified as high on PC using a median split) indicate that they would feel greater distance from relational partners upon being criticized by them and that relational distancing was greatest in the context of relationships the authors assume to be closer (e.g., family vs. workplace relationships). I generally believe that these findings are interesting. However, the manuscript has a number of significant issues, including problems with the conceptualization of PC and other constructs, the presentation of findings, an unclear approach to selecting and presenting relevant background literature, and analytic limitations. It is possible that many of these concerns could be addressed in a major revision.

My primary concerns surround 1) the conceptualization and interpretation of PC and other constructs; 2) the appropriate selection and representation of background literature; and 3) the analytic approach.

Thank you for your comments. Please find our responses to your comments below. 

1. Concerns about the conceptualization and interpretation of PC and other constructs

a. The first sentence of the introduction reduces criticism to a necessarily negative experience, yet we know that this is not always the case—criticism can be experienced positively, and much of the impact of criticism may rely on attributions about criticism (see e.g., Allred & Chambless, 2014; Peterson et al., 2009; Chambless et al., 2010). The second sentence also suggests that criticism is necessarily a form of social rejection, which we know is not the case. Oftentimes, critical relatives do not mean to be rejecting but may have an internal locus of control that guides them to believe that criticism will help their family member to change (see Hooley’s 2007 review of expressed emotion). People receiving criticism may also make positive attributions about it (e.g., “my mother is saying this to try and help me”). The authors do consider attributions in the next section, but that body of work needs to inform their introduction paragraph as well.

Thank you for your comment. We have edited the introduction to reflect the possibility of criticism in being construed as either a positive or negative experience and included information on attributions in the introduction paragraph as follows: 

“Criticism, which is defined as negative evaluative feedback received from other people in social interactions [1-2]. It can be construed by recipients as either a positive or distressing experience, where much of the impact of criticism is contingent on the attributions the individual makes about the criticism [3-5]. Receiving social evaluations such as social rejection in the form of negative feedback can result in negative self-evaluations [6] and social pain [7]. Criticism has been reported to be involved in majority of descriptions of hurtful events [8] and makes up a category of hurtful interactions in romantic relationships [9]. On the other hand, positive attributions can also be made about criticism such as constructive criticism, which can be interpreted as promoting understanding and providing direction for improvement [10]. Hence, the investigation of individual differences in the perception of criticism in interpersonal relationships is important in providing insight to understanding the negative consequences of criticism on relationships.”

b. p. 4 lines 74-76. This statement is not exactly supported or in line with how the authors measure criticism. The PCM does NOT actually assess frequency of criticism at all, but rather the global perception of criticism from a relationship partner, which could be influenced by a number of factors which include but which are certainly not limited to frequency of criticism. If the authors want to test the idea that “an individual who has a greater tendency to perceive criticism in their interpersonal relationships would experience greater feelings of hurt in response to criticism,” then they need to use a different measure.

Thank you for pointing this out. Yes, we agree that the PCM does not assess frequency of criticism, but a global perception of criticism from a relationship partner. We have clarified our meaning in this sentence as follows: “Hence, it can be expected that an individual who perceives their relationship partner to be more critical of them would be more likely to experience greater feelings of hurt.”

c. p. 4 lines 78-79. The statement that the PCM “usually” reflects “perceptions of destructive criticism” is not a fair characterization of that finding.

We have edited this statement to clarify the finding: “One measure of the perception of criticism in one’s relationships is the Perceived Criticism Measure (PC) [12], which reflects the amount of criticism in an individual’s closest or most meaningful relationships, such as with a romantic partner, spouse, or parent and shows a strong correlation with perceptions of destructive criticism [13].”

d. p. 6 lines 121-126: this paragraph seems like very weak justification for proposing differences across different relationships/sources. The authors may do better to use the PC literature to support the idea that source matters (e.g., it is highly surprising that Renshaw, 2008 is not considered). The material on self-concept does not seem to add much to the author’s presentation of background and feels somewhat disjointed.

We have removed the part on self-concept and included a discussion incorporating PC literature as follows: “Mental representations of one’s relationships with others are expected to inform an individual’s behaviour while being used to predict and interpret others’ behaviours [21], suggesting that the same criticism may elicit different reactions when originating from different sources. The relationship of PC with depressive symptoms also differs across sources where only PC ratings for family members and romantic partners who lived with the respondents significantly predicted change in depressive symptoms but PC ratings for friends and individuals ranked most influential did not [22]. This finding suggests that the impact of perceived criticism on an individual may differ depending on the environment in which the relationship is embedded in. In addition, individuals who are depressed or maritally discordant have also been found to display a criticality bias – a tendency to over-perceive criticism – which was found to be associated with marital attributions of behaviour [23], suggesting that such bias appears to be indicative of the views an individual holds of their spouse and marriage.”

e. The authors seem to assume differential levels of closeness between different relationship partners. However, closeness cannot be assumed just based on relationship and should likely have been measured in this study. Do the authors have data on closeness? Or data to speak to whether participants lived with (i.e., were at least physically close to) certain partners (see Renshaw, 2008)?

Thank you for your comment. We have included relationship quality as a covariate in the analysis and the pattern of the results remain unchanged. 

f. p. 12 line 244: the authors seem to be misusing the term “individual factors” – they did not actually test individual factors (e.g., personality variables, other individual differences) that predict hurt and relational distancing besides hurt and relational distancing.

We have corrected this as follows: “The present study aimed to investigate the relationship between perceptions of criticism from relationship partners in influencing the response to criticism in different relational contexts in terms of the experiences of hurt and relational distancing.”

g. The finding that the high PC group did not report higher levels of hurt is consistent with past work that is not mentioned here (e.g., Masland, Drabu, & Hooley, 2018)

Thank you for pointing this out. We have included this in our discussion as follows: “The finding that the high PC group did not report feeling higher levels of hurt is consistent with previous findings that PC was not significantly correlated with criticism sensitivity, which includes emotional sensitivity – measured by how upset an individual is by criticism [29].”

h. The authors note that their findings, which are not “behavioral observations” as they describe them (p. 13 line 249) are discordant with neuroimaging findings. This needs expansion and further explanation/interpretation

We have replaced the term “behavioral observations” with “findings” and elaborated on this point as follows: “These findings appear at first glance to differ from those in previous neuroimaging results discussed earlier [15]. One possible interpretation of these findings could be that participants with high PC employed relational distancing as a coping mechanism by disengaging themselves from the hurt and social pain associated with criticism. This possibility is consistent with past work on how individuals may use of ego-moving perspective of time, which enables people to psychologically distance themselves from unpleasant, threatening past experiences [40]. Since rejection from close others tends to hurt more than rejection from strangers [8, 26], it could be that a sense of increased distance from one’s relational partners can reduce the intensity of hurt experienced in response to hurtful behaviour such as criticism. Hence, it is possible that participants in the high PC group tended to distance themselves from the relationship and in doing so, “distanced” their emotions from the situation and did not show significantly higher levels of hurt feelings compared to participants in the low PC group. In terms of the neural activation patterns found in previous studies, such a coping mechanism where individuals distance themselves from the relationship and the associated emotions resulting from the criticism may be reflected in the decreased activity previously observed in cognitive control networks as less attention and cognitive resources are being placed on processing the criticism. It is possible that as Hooley et al. [15] found in terms of increased amygdala activity, participants in the high PC group may have experienced more hurt towards the criticism described in the vignettes but did not rate their levels of hurt as significantly higher than participants due to the employment of relational distancing in order to cope with the hurt that they experienced.”

i. How does the authors’ interpretations of the possible roles of distancing and habituation (p. 13) square with the decades of research showing that high PC predicts poor clinical outcomes? Or with neuroimaging findings? For example, habituation does not seem consistent with DLPFC findings (Hooley, Siegle, & Gruber, 2012) or the results of cognitive work (Masland et al., 2015)

We have elaborated on this point as follows: “Another possible interpretation is that participants who tend to perceive higher levels of criticism in a relationship showed emotional habituation towards the experience of hurt arising from criticism, such that the more a participant perceives a relationship counterpart to be critical of them, the more accustomed they were to criticism from these relational partners. This could lead to less hurt experienced with repeated encounters involving criticism. On the other hand, previous findings have indicated that PC reliably predicts clinical outcomes [see 14 for review] and greater subjective distress experienced by bipolar disorder patients in response to familial criticism predicted the severity of depressive and manic symptoms [36]. A possible explanation for present findings in the context of previous work could be the association between attributions for the criticism and PC. Chambless et al. [5] found that higher scores for negative attributions predicted higher PC ratings while Peterson et al. [41] found similar associations between PC ratings and the cause and responsibility ascribed to partners’ behaviours. It is possible that despite habituating to the experience of hurt from their partners, high PC individuals are more likely to perceive “more” criticism from their partners as well as form negative attributions about such criticisms along with a majority of their partners’ behaviours in general. It could be that these high PC individuals experience higher levels of hurt from a consistent perception of multiple events of negative behaviour from their relationship partners as opposed to significantly higher levels of hurt from a single event, compared to low PC individuals who may be less likely to perceive and explain behaviours from their relationship partners as negative. A more consistent perception of negative behaviours from relationship partners could then contribute to the degradation of their relationships with their partners and consequently, to poor clinical outcomes as well. In addition, greater relational distancing observed in participants with high PC also suggest a possible pathway through which PC ratings predict clinical outcomes. By distancing themselves from the relationship partner, high PC individuals may experience greater social isolation or be less likely to seek social support from these relationships, leading to reduced relationship satisfaction which could consequently influence clinical outcomes. Further research can be conducted in order to link the experience of relational distancing with relationship outcomes and clinical outcomes.” 

We have also raised possible explanations for how results may be related to previous neuroimaging findings as follows: “One possible interpretation of these findings could be that participants with high PC employed relational distancing as a coping mechanism by disengaging themselves from the hurt and social pain associated with criticism. This possibility is consistent with past work on how individuals may use of ego-moving perspective of time, which enables people to psychologically distance themselves from unpleasant, threatening past experiences [40]. Since rejection from close others tends to hurt more than rejection from strangers [8, 26], it could be that a sense of increased distance from one’s relational partners can reduce the intensity of hurt experienced in response to hurtful behaviour such as criticism. Hence, it is possible that participants in the high PC group tended to distance themselves from the relationship and in doing so, “distanced” their emotions from the situation and did not show significantly higher levels of hurt feelings compared to participants in the low PC group. In terms of the neural activation patterns found in previous studies, such a coping mechanism where individuals distance themselves from the relationship and the associated emotions resulting from the criticism may be reflected in the decreased activity previously observed in cognitive control networks as less attention and cognitive resources are being placed on processing the criticism. It is possible that as Hooley et al. [15] found in terms of increased amygdala activity, participants in the high PC group may have experienced more hurt towards the criticism described in the vignettes but did not rate their levels of hurt as significantly higher than participants due to the employment of relational distancing in order to cope with the hurt that they experienced.”

j. p. 15 line 302: the authors suggest that workplace relationships are different from romantic/family relationships because they are “temporary or… by choice.” However, romantic relationships can also be temporary or by choice. This is a good example of why it is problematic that the authors assumed closeness in certain relationships without actually measuring the degree of closeness

Thank you for your comment. Please see our response above to the inclusion of relationship quality in the analysis.

k. p. 15 line 308: the authors again assume closeness although it was not measured. They also do not seem to recognize that attributions about PC vary even in close relationships (e.g., Allred & Chambless, 2014; Peterson et al., 2009; Chambless et al., 2010)

We have included this in a discussion on the attributions of criticism as follows: “Mental representations of one’s relationships with others are expected to inform an individual’s behaviour while being used to predict and interpret others’ behaviours [18], suggesting that the same criticism may elicit different reactions when originating from different sources. The relationship of PC with depressive symptoms also differs across sources where only PC ratings for family members and romantic partners who lived with the respondents significantly predicted change in depressive symptoms but PC ratings for friends and individuals ranked most influential did not [23]. This finding suggests that the impact of perceived criticism on an individual may differ depending on the environment in which the relationship is embedded in. In addition, individuals who are depressed or maritally discordant have also been found to display a criticality bias – a tendency to over-perceive criticism – which was found to be associated with marital attributions of behaviour [23], suggesting that such bias appears to be indicative of the views an individual holds of their spouse and marriage.”

l. Some research has examined emotional upset to criticism, including Miklowitz et al. (2005), who found that emotional upset, rather than criticism, predicted outcomes for bipolar disorder. I am quite surprised that this is not considered given the manuscript’s focus on hurt

We have included the research by Miklowitz et al. (2005) in the introduction and discussion as follows: “Given the well-established empirical association between excessive criticism and levels of PC with psychopathology [14, 35; see 14 for a review] and findings that ratings of emotional upset in response to relatives’ criticism predicted depressive and manic symptoms in bipolar patients [36], studying how PC relates to feelings of hurt and relational distancing can provide insight into the relationship between an individual’s perception of criticism and the consequences on emotions and the relationship.” 

m. p. 16 line 336: the authors suggest that their findings show “that considerations of individual differences in how criticism is perceived are important for workplace supervisors in building healthy relationships and motivating employees” yet they did not measure individual differences in how PC is perceived. They measured the influence of PC and source on relational distancing and hurt, not what individual differences might predict PC

Thank you for bringing this to our attention. We have clarified our phrasing in this line as follows: “Findings suggest that considerations of perceived criticism are important for workplace supervisors in building healthy relationships and motivating employees.”

n. It is not enough to say that there are cultural differences of note. I would like to see more about how specifically these differences may have limited or influenced the results

In response to comments from both reviewers, we have elaborated further on these cultural differences in the introduction and discussion as follows: 

“Few studies have also investigated the PC construct in Asian contexts [e.g. 30-31] although cultural differences in the perception and attributions of criticism have also been found [e.g. 10, 31]. Previous research indicated (i) correlations between patients’ perceived criticism and observer ratings of criticism from relatives only in White patients but not Black patients [32], (ii) observer ratings of criticism predicted relapse and poor clinical outcomes only in White but not Black participants and (iii) associations between perceived criticism and poor outcomes in both groups [33-34]. More specifically, Black participants in a community sample reported more positive attributions but perceived greater destructive criticism compared to White participants [3]. Allred & Chambless [10] also found that Black participants were significantly less upset by perceived criticism from relatives compared to White participants and perceptions of relatives’ warmth were only observed to be correlated with less upset for Black participants and not White participants.”

“Expression of hurt feelings may be curtailed in collectivist cultures to preserve group harmony or conform with group values whereas individualistic cultures tend to promote self-expression and expression of feelings [45]. Collectivistic cultures also place a greater emphasis on hierarchy and status, suggesting that the extent of hurt feelings and relational distancing may differ in response to sources of criticism in positions of higher status, such as parents and workplace supervisors. In addition, collectivistic cultures tend to employ emotion suppression as an emotion regulation strategy as it minimises the risk of disrupting group harmony whereas individualistic cultures tend to use cognitive reappraisal [57]. As a result of these cultural tendencies, in the context of Singapore where the present study was conducted, criticism from figures of authority such as supervisors or parents may be more common and participants may be less likely to feel hurt or distanced in response to criticism occurring in these relationships. As discussed in the introduction, perceptions and attributions of criticism have been found to differ across cultures [3, 10, 31]. Hence, similar to these findings in [3, 10], cultural differences in the attributions and perceptions of warmth may be a possible explanation for the findings in the present study between PC ratings and feelings of hurt and relational distancing.” 

2. Appropriate selection and representation of background literature: generally I am concerned that the authors have not adequately tapped into the somewhat small but very rich PC literature. They do not seem to consistently cite the most relevant research.

a. p. 7 line 154: there is a wealth of literature that supports their claim of a well-established link between PC and psychopathology, and it is unclear why they chose the two studies they did in this location, particularly #24. They could cite a review paper here or could cite a broader range of relevant and seminal literature supporting this link (e.g., see papers reviewed in Masland & Hooley, 2015)

We have removed Citation #24 and replaced it with the Masland & Hooley (2015) review for readers to refer to as follows: “Given the well-established empirical association between excessive criticism and levels of PC with psychopathology [14, 35; see 14 for a review] and findings that ratings of emotional upset in response to relatives’ criticism predicted depressive and manic symptoms in bipolar patients [36], studying how PC relates to feelings of hurt and relational distancing can provide insight into the relationship between an individual’s perception of criticism and the consequences on emotions and the relationship.”

b. p. 7 line 150: this may be true, but there has been some limited work on criticism sensitivity that is relevant here (e.g., Masland et al., 2018; White et al., 1998)

Thank you for pointing this out. We have included mention of some of this relevant work as follows: “A limited number of studies have investigated sensitivity to criticism where criticism sensitivity was found to exhibit convergent validity with measures of upset in response to criticism [White et al., 1998] but not with perceptions of criticism [White et al., 1998; Masland et al., 2019]. However, previous studies have not looked specifically at (i) how individual differences in the perception and response to criticism influence experiences of hurt and relational distancing as a result of criticism and (ii) whether emotional sensitization or habituation occurs in response to criticism in various interpersonal relationships.”

c. Where the authors use citation #31 it would be more appropriate to cite work related to attributions and PC (e.g., e.g., Allred & Chambless, 2014; Peterson et al., 2009; Chambless et al., 2010)

We have included an elaboration citing work related to attributions and PC as follows: “In addition, previous work has found an association between attributions and perceptions of criticism. Specifically, higher negative attribution scores were related to higher PC ratings [3, 5], indicating that attributions are related to how much criticism is perceived in a relationship. Similarly, [41] found that self-reports of causal and responsibility attributions for negative spousal behaviour were related to all types of criticism, suggesting that the attributions individuals make about another’s behaviour influence whether the behaviour will be perceived or interpreted to be criticism. Hence, it is possible that individuals may be are more likely to make negative attributions such as being more likely to attribute the cause of the negative feedback received to negative dispositions of the supervisor as opposed to family members, with whom individuals have a longer shared history and knowledge of and a more permanent relationship.” 

d. Citation 19 does not seem to apply necessarily to criticism but more broadly to hurtful interactions. This manuscript would be much better support and contextualized with more reliance on the PC literature

We have included more support in the manuscript with citations from the PC literature in the manuscript in the following instances: 

“Mental representations of one’s relationships with others are expected to inform an individual’s behaviour while being used to predict and interpret others’ behaviours [21], suggesting that the same criticism may elicit different reactions when originating from different sources. The relationship of PC with depressive symptoms also differs across sources where only PC ratings for family members and romantic partners who lived with the respondents significantly predicted change in depressive symptoms but PC ratings for friends and individuals ranked most influential did not [22]. This finding suggests that the impact of perceived criticism on an individual may differ depending on the environment in which the relationship is embedded in. In addition, individuals who are depressed or maritally discordant have also been found to display a criticality bias – a tendency to over-perceive criticism – which was found to be associated with marital attributions of behaviour [23], suggesting that such bias appears to be indicative of the views an individual holds of their spouse and marriage.”

 “In addition, previous work has found an association between attributions and perceptions of criticism. Specifically, higher negative attribution scores were related to higher PC ratings [3, 5], indicating that attributions are related to how much criticism is perceived in a relationship. Similarly, [41] found that self-reports of causal and responsibility attributions for negative spousal behaviour were related to all types of criticism, suggesting that the attributions individuals make about another’s behaviour influence whether the behaviour will be perceived or interpreted to be criticism. Hence, it is possible that individuals may be are more likely to make negative attributions such as being more likely to attribute the cause of the negative feedback received to negative dispositions of the supervisor as opposed to family members, with whom individuals have a longer shared history and knowledge of and a more permanent relationship.” 

“Few studies have also investigated the PC construct in Asian contexts [e.g. 30-31] although cultural differences in the perception and attributions of criticism have also been found [e.g. 10, 31]. Previous research indicated (i) correlations between patients’ perceived criticism and observer ratings of criticism from relatives only in White patients but not Black patients [32], (ii) observer ratings of criticism predicted relapse and poor clinical outcomes only in White but not Black participants and (iii) associations between perceived criticism and poor outcomes in both groups [33-34]. More specifically, Black participants in a community sample reported more positive attributions but perceived greater destructive criticism compared to White participants [3]. Allred & Chambless [10] also found that Black participants were significantly less upset by perceived criticism from relatives compared to White participants and perceptions of relatives’ warmth were only observed to be correlated with less upset for Black participants and not White participants.”

e. The authors do not seem to recognize that attributions about PC vary even in close relationships (e.g., Allred & Chambless, 2014; Peterson et al., 2009; Chambless et al., 2010)

We have included this in a discussion on the attributions of criticism as follows: “Mental representations of one’s relationships with others are expected to inform an individual’s behaviour while being used to predict and interpret others’ behaviours [21], suggesting that the same criticism may elicit different reactions when originating from different sources. The relationship of PC with depressive symptoms also differs across sources where only PC ratings for family members and romantic partners who lived with the respondents significantly predicted change in depressive symptoms but PC ratings for friends and individuals ranked most influential did not [22]. This finding suggests that the impact of perceived criticism on an individual may differ depending on the environment in which the relationship is embedded in. In addition, individuals who are depressed or maritally discordant have also been found to display a criticality bias – a tendency to over-perceive criticism – which was found to be associated with marital attributions of behaviour [23], suggesting that such bias appears to be indicative of the views an individual holds of their spouse and marriage.”

f. Some research has examined emotional upset to criticism, including Miklowitz et al. (2005), who found that emotional upset, rather than criticism, predicted outcomes for bipolar disorder. I am quite surprised that this is not considered given the manuscript’s focus on hurt

We have included the research by Miklowitz et al. (2005) in the introduction and discussion as follows: “Given the well-established empirical association between excessive criticism and levels of PC with psychopathology [14, 35; see 14 for a review] and findings that ratings of emotional upset in response to relatives’ criticism predicted depressive and manic symptoms in bipolar patients [36], studying how PC relates to feelings of hurt and relational distancing can provide insight into the relationship between an individual’s perception of criticism and the consequences on emotions and the relationship.”

3. Analytic approach

a. The link for the data repository appears to be broken

Thank you or pointing this out. We have fixed the link. 

b. Why did the authors use a median split for PC ratings? I am aware that this is very common in the PC literature, but it nevertheless requires justification as it has significant limitations as an analytic strategy. In this case it seems that a dimensional approach is both possible and likely to be more informative

Thank you for your comment. As mentioned, we initially used the median split as previous studies have done (e.g. Hooley et al., 2012). Following your suggestion, we have now updated the data analysis to use a dimensional approach instead and the same pattern of results emerged.

c. There are a number of analyses missing that should be included for complete review of this paper and its findings: the range of PC scores (including the range in each group), the overall PC mean/SD and means/SDs by group, the correlation of PC with distancing and hurt, the correlation of distancing and hurt

We have included: (i) the range of PC scores, (ii) the correlations between PC with hurt and distancing and (iii) the correlation between hurt and distancing in the Results section. The correlations are summarised in Table 2.

d. Although the authors use a categorical analysis approach, they inappropriately use dimensional language to describe their findings (e.g., the abstract reads “the more critical participants perceived the relational partner to be, the more distanced they felt upon receiving criticism from them”)

Thank you for pointing this out. As mentioned above, we have updated the data analysis to use a dimensional approach. 

4. Additional Concerns

a. The authors describe sensitization and habituation models for understanding the impact of criticism. Their hypotheses align with a sensitization model, but they do not give sufficient justification for why they chose the sensitization model over the habituation model

We have elaborated further on the hypotheses as follows: “In line with previous findings on the association between PC and (i) increased upset [10, 36-37] and (ii) lower relationship and marital satisfaction [23, 38], we formulated the following hypotheses: 

Hypothesis 1: Individuals who have higher perceived criticism ratings of their relational partner would experience higher levels of hurt and relational distancing in response to criticism compared to individuals who have lower perceived criticism ratings. 

Hypothesis 2: Higher levels of hurt and lower relational distancing will be experienced in familial relationships compared to other social relationships.”

b. On a more granular level, there are times when the manuscript is difficult to follow because of missing words or sentence structures that could be more straightforward. This is a minor concern in the broader context of this review.

We have looked through the manuscript and made the necessary revisions. 

 

Reviewer #2: This is interesting research in a novel area. The paper is generally well written. The statistics are simple but effectively examine the research questions. Some minor changes are recommended below. In particular, some additional statistics are required. Overall, this is valuable research that adds to the current knowledge-base.

Thank you for your comments. Please see our responses to your comments below. 

Abstract

The first two sentences of abstract are confusing. Please re-phrase

We have edited the first two sentences as follows: “Criticism is commonly perceived as hurtful and individuals may respond differently to criticism originating from different sources. However, the influence of an individual’s perception of criticism in their social relationships on negative emotional reactions to criticism across different relational contexts has not been examined across different relational contexts.”

Introduction

Line 153 mentions that PC is linked to psychopathology. It would be beneficial, at some point in the introduction, to briefly mention which psychopathologies/diagnoses specifically are linked to PC to clarify the clinical relevance of examining PC

We have included mention of the specific psychopathologies that have been associated with PC as follows: “Given the well-established empirical association between excessive criticism and levels of PC with psychopathology [14, 35; see 14 for a review] and findings that ratings of emotional upset in response to relatives’ criticism predicted depressive and manic symptoms in bipolar patients [36], studying how PC relates to feelings of hurt and relational distancing can provide insight into the relationship between an individual’s perception of criticism and the consequences on emotions and the relationship.”

Method

Information should be included indicating where participants were recruited from (i.e. country)

We have included this information in the Methods section as follows: “Participants were recruited in Singapore (N = 178, male = 83, female = 95, Mage = 21.3, SDage = 2.23) through (i) a psychology undergraduate course and compensated with course credits and (ii) advertisements and compensated with remuneration (Table 1).

Results

Were there any differences in results between university-recruited cohort and the other cohort? This should be assessed statistically

Thank you for your comment. We have included a preliminary analysis where we found that there were no significant differences between the two groups in their ratings of (i) hurt and (ii) relational distancing as follows: “We conducted independent samples t-test to check whether there were differences between the participants recruited from the university and those who were not. There were no significant differences between the two groups in terms of (i) ratings of hurt (t(621) = 0.19, p = 0.85) and (ii) ratings of relational distancing (t(617) = -0.09, p = 0.93) . Hence, both groups were analysed together in the main analyses.”

Was there any significant difference in results for participants who were working/previously worked vs never working? Significant differences between those who were in/had been in relationship vs never been in relationship? Again analysis is needed to test this.

The inclusion of 8 participants who had never worked is problematic, as they would never have experienced a professional supervisor, as is the inclusion of participants who have never been in a relationship

Thank you for your clarification. The ratings for the vignettes of participants who have never been in a romantic relationship and those who did not have previous working experience were excluded. For example, the rating for the vignette describing criticism from a romantic partner made by a participant who indicated that they have never been in a romantic relationship will not be included in the analysis We have included this clarification in the Analytic Plan as follows: “For participants who indicated that they (i) have never been in a romantic relationship and/or (ii) did not have previous work experience, their ratings for the vignettes involving criticism from (i) romantic partners and (ii) supervisors respectively were not included in the data analysis.”

Discussion

Line 269 “Secondly, results in the present study partially supported Hypothesis 2 where relational distancing was significantly across the different relationships but levels of hurt were not.” This is unclear. Please rephrase.

We have rephrased the sentence to clarify our meaning as follows: “Secondly, results in the present study only partially supported Hypothesis 2, where we hypothesised that higher levels of hurt and relational distancing would be observed in familial relationships compared to other social relationships. However, in the present study, only significant differences in levels of relational distancing but not hurt were observed across different relationships.”

Line 271 “Results suggest that hurt feelings…” Also unclear, please re-phrased

We have rephrased the sentence to clarify our meaning as follows: “Results suggest that hurt feelings may not differ in response to criticism originating from the different relationships that were examined in this study. Rather, findings suggest that criticism can be hurtful as long as the individual receiving it perceives it to be.”

The discussion includes examination of the possible implication of cultural contexts. This is an important point. However, it is not included at all in the introduction. It would be beneficial to have the review of previous research indicate in which cultural contexts previous research was conducted and greater discussion in the introduction about this issue. Further, the aim of the research should be amended, i.e. aim: to examine the research questions in the context of the Singaporean culture.

In response to comments from both reviewers, we have elaborated further on these cultural differences in the introduction and discussion as follows: 

“Few studies have also investigated the PC construct in Asian contexts [e.g. 30-31] although cultural differences in the perception and attributions of criticism have also been found [e.g. 10, 31]. Previous research indicated (i) correlations between patients’ perceived criticism and observer ratings of criticism from relatives only in White patients but not Black patients [32], (ii) observer ratings of criticism predicted relapse and poor clinical outcomes only in White but not Black participants and (iii) associations between perceived criticism and poor outcomes in both groups [33-34]. More specifically, Black participants in a community sample reported more positive attributions but perceived greater destructive criticism compared to White participants [3]. Allred & Chambless [10] also found that Black participants were significantly less upset by perceived criticism from relatives compared to White participants and perceptions of relatives’ warmth were only observed to be correlated with less upset for Black participants and not White participants.”

“Expression of hurt feelings may be curtailed in collectivist cultures to preserve group harmony or conform with group values whereas individualistic cultures tend to promote self-expression and expression of feelings [45]. Collectivistic cultures also place a greater emphasis on hierarchy and status, suggesting that the extent of hurt feelings and relational distancing may differ in response to sources of criticism in positions of higher status, such as parents and workplace supervisors. In addition, collectivistic cultures tend to employ emotion suppression as an emotion regulation strategy as it minimises the risk of disrupting group harmony whereas individualistic cultures tend to use cognitive reappraisal [57]. As a result of these cultural tendencies, in the context of Singapore where the present study was conducted, criticism from figures of authority such as supervisors or parents may be more common and participants may be less likely to feel hurt or distanced in response to criticism occurring in these relationships. As discussed in the introduction, perceptions and attributions of criticism have been found to differ across cultures [3, 10, 31]. Hence, similar to these findings in [3, 10], cultural differences in the attributions and perceptions of warmth may be a possible explanation for the findings in the present study between PC ratings and feelings of hurt and relational distancing.” 

We have also edited the aim of the research to clarify the Singaporean context of the present study: "Few studies have also investigated the PC construct in Asian contexts [e.g. 30-31] although cultural differences in the perception and attributions of criticism have also been found [e.g. 10, 31]. Previous research indicated (i) correlations between patients’ perceived criticism and observer ratings of criticism from relatives only in White patients but not Black patients [32], (ii) observer ratings of criticism predicted relapse and poor clinical outcomes only in White but not Black participants and (iii) associations between perceived criticism and poor outcomes in both groups [33-34]. More specifically, Black participants in a community sample reported more positive attributions but perceived greater destructive criticism compared to White participants [3]. Allred & Chambless [10] also found that Black participants were significantly less upset by perceived criticism from relatives compared to White participants and perceptions of relatives’ warmth were only observed to be correlated with less upset for Black participants and not White participants.

Given the well-established empirical association between excessive criticism and levels of PC with psychopathology [14, 35; see 14 for a review] and findings that ratings of emotional upset in response to relatives’ criticism predicted depressive and manic symptoms in bipolar patients [36], studying how PC relates to feelings of hurt and relational distancing can provide insight into the relationship between an individual’s perception of criticism and the consequences on emotions and the relationship. Hence, the present study aims to investigate the relationship between individual differences in PC and experiences of hurt and relational distancing in response to criticism in four different relational contexts: (i) romantic partners, (ii) mothers, (iii) fathers, and (iv) workplace supervisors in a Singaporean sample.”

A large proportion of the sample had never been in a romantic relationship. Given that one component of the research was about romantic relationships, this is a notable limitation of the research and should be mentioned in the discussion in the ‘limitations’ section.

We have included this in the limitations section as follows: “In addition, a large proportion of the sample in the present study had never been in a romantic relationship. Hence, future studies can look further into criticism occurring in romantic relationships such as criticism between romantic partners as well as spouses.”

---

## [Decision Letter · Decision Letter 1]

21 Jun 2022

PONE-D-21-36615R1Negative emotional reactions to criticism: Perceived criticism and source affects extent of hurt and relational distancingPLOS ONE

Dear Dr. Esposito,

Thank you for submitting your manuscript to PLOS ONE. After careful consideration, we feel that it has merit but does not fully meet PLOS ONE’s publication criteria as it currently stands. Therefore, we invite you to submit a revised version of the manuscript that addresses the points raised during the review process.

Reviewer 2 is satisfied with the edits to your manuscript, but reviewer 1 has remaining concerns that they would like to have addressed.

We look forward to receiving your revised manuscript.

Kind regards,

Sarah Hope Lincoln

Academic Editor

PLOS ONE

Reviewers' comments:

Reviewer's Responses to Questions

**Comments to the Author**

1. If the authors have adequately addressed your comments raised in a previous round of review and you feel that this manuscript is now acceptable for publication, you may indicate that here to bypass the “Comments to the Author” section, enter your conflict of interest statement in the “Confidential to Editor” section, and submit your "Accept" recommendation.

Reviewer #1: (No Response)

Reviewer #2: All comments have been addressed

2. Is the manuscript technically sound, and do the data support the conclusions?

Reviewer #1: Partly

Reviewer #2: Yes

3. Has the statistical analysis been performed appropriately and rigorously? 

Reviewer #1: No

Reviewer #2: Yes

4. Have the authors made all data underlying the findings in their manuscript fully available?

Reviewer #1: Yes

Reviewer #2: Yes

5. Is the manuscript presented in an intelligible fashion and written in standard English?

Reviewer #1: Yes

Reviewer #2: Yes

6. Review Comments to the Author

Reviewer #1: My initial review was extensive, and the authors have done a commendable job of addressing my concerns. However, two significant concerns remain:

1. In response to my comments about assuming closeness based on type of relationship, the authors noted that they controlled for relationship quality. However, relationship quality is not the same as closeness. I could have a really fantastic (high quality) relationship with my mailman but not be very close to him. Similarly, I could have a really terrible relationship (low quality) with my sister, but also be very close to her (e.g., situations of abuse, co-dependency, enmeshment). Controlling for relationship quality does not address the issues of assuming closeness based on relationship type. Relatedly, the authors do not include any information about how they measured relationship quality, which is a significant omission.

2. In response to my comments about using a median split, the authors changed their analytic approach. I appreciate this effort (although I could have been easily convinced that a median split was appropriate given the precedent in the PC literature—I was mostly suggesting greater justification). However, the analysis is now confusing and seems disjointed. They still use F statistics and refer to PC groups (e.g., Tables 2a and 2b and the associated text). Table 4 also still refers to low and high PC groups.

Reviewer #2: The authors have appropriately responded to reviewer comments and edited the manuscript accordingly.

7. PLOS authors have the option to publish the peer review history of their article (what does this mean?). If published, this will include your full peer review and any attached files.

Reviewer #1: No

Reviewer #2: No

---

## [Author Response · Author response to Decision Letter 1]

22 Jun 2022

Reviewer #1: My initial review was extensive, and the authors have done a commendable job of addressing my concerns. However, two significant concerns remain:

Thank you for your comments. Please find our responses to your comments below:

1. In response to my comments about assuming closeness based on type of relationship, the authors noted that they controlled for relationship quality. However, relationship quality is not the same as closeness. I could have a really fantastic (high quality) relationship with my mailman but not be very close to him. Similarly, I could have a really terrible relationship (low quality) with my sister, but also be very close to her (e.g., situations of abuse, co-dependency, enmeshment). Controlling for relationship quality does not address the issues of assuming closeness based on relationship type. Relatedly, the authors do not include any information about how they measured relationship quality, which is a significant omission.

We have included information on how relationship quality was measured as follows: “Relationship quality was measured using questions adapted from the Quality of Marriage Index [40]. Participants rated the quality of each of the relationship type: romantic partner, mother, father and workplace supervisor. There were 5 items rated on a 7-point Likert scale and 1 item rated on a 10-point Likert scale where higher scores reflected higher relationship quality.”

We agree that relationship quality and relationship closeness are different and we acknowledge the assumption of closeness based on relationship type in the limitations as follows: “The analysis of the present study was also assumed relationship closeness based on the relationship type where familial and intimate relationships were assumed to be close compared to the relationship with a supervisor. Future studies can include measures of relationship closeness to examine whether experiences of relationship closeness also play a role in the response towards criticism originating in different relationship types.”

2. In response to my comments about using a median split, the authors changed their analytic approach. I appreciate this effort (although I could have been easily convinced that a median split was appropriate given the precedent in the PC literature—I was mostly suggesting greater justification). However, the analysis is now confusing and seems disjointed. They still use F statistics and refer to PC groups (e.g., Tables 2a and 2b and the associated text). Table 4 also still refers to low and high PC groups.

Thank you for your comment. We have used the median split for PC and updated the manuscript accordingly. 

We have also cited the use of the median split in previous PC literature as follows: “Participants were grouped into high or low on PC through a median split (MdnMother = 4, MdnFather, MdnPartner, MdnSupervisor = 5). The medians in this sample are also similar to previous studies on PC which have also used median splits (e.g. [12] and [41]) to facilitate interpretation of results.”

---

## [Decision Letter · Decision Letter 2]

10 Jul 2022

Negative emotional reactions to criticism: Perceived criticism and source affects extent of hurt and relational distancing

PONE-D-21-36615R2

Dear Dr. Esposito,

We’re pleased to inform you that your manuscript has been judged scientifically suitable for publication and will be formally accepted for publication once it meets all outstanding technical requirements.

Kind regards,

Sarah Hope Lincoln

Academic Editor

PLOS ONE

Additional Editor Comments (optional):

Reviewers' comments:

Reviewer's Responses to Questions

**Comments to the Author**

1. If the authors have adequately addressed your comments raised in a previous round of review and you feel that this manuscript is now acceptable for publication, you may indicate that here to bypass the “Comments to the Author” section, enter your conflict of interest statement in the “Confidential to Editor” section, and submit your "Accept" recommendation.

Reviewer #1: All comments have been addressed

2. Is the manuscript technically sound, and do the data support the conclusions?

Reviewer #1: Yes

3. Has the statistical analysis been performed appropriately and rigorously? 

Reviewer #1: Yes

4. Have the authors made all data underlying the findings in their manuscript fully available?

Reviewer #1: Yes

5. Is the manuscript presented in an intelligible fashion and written in standard English?

Reviewer #1: Yes

6. Review Comments to the Author

Reviewer #1: Although I remain concerned about the conflation of closeness and relationship type, the authors have revised the manuscript to include some discussion of this limitation and I have no further feedback.

7. PLOS authors have the option to publish the peer review history of their article (what does this mean?). If published, this will include your full peer review and any attached files.

Reviewer #1: No

---

## [Editor Report · Acceptance letter]

29 Jul 2022

PONE-D-21-36615R2 

Negative emotional reactions to criticism: Perceived criticism and source affects extent of hurt and relational distancing 

Dear Dr. Esposito:

I'm pleased to inform you that your manuscript has been deemed suitable for publication in PLOS ONE. Congratulations! Your manuscript is now with our production department. 

Kind regards, 

on behalf of

Dr. Sarah Hope Lincoln 

Academic Editor

PLOS ONE